# Study on Dyeing Properties and Color Characteristics of Wool Fabrics Dyed with *Geranium caespitosum* L. Extract—A New Natural Yellow Dye

Zhijun Zhao [1,*]  , Chunxiao Yan [1], Fei Xu [1] and Jianhong Liu [2]

1   Department of Clothing and Costume Design, College of Art & Design, Qiqihar University, Qiqihar 161006, China; 2021936796@qqhru.edu.cn (C.Y.); 02004@qqhru.edu.cn (F.X.)
2   Department of Mechanical and Electrical Engineering, Qiqihar School of Engineering, Qiqihar 161005, China
*   Correspondence: 02017@qqhru.edu.cn

**Abstract:** Natural dyes play an important role in sustainable dyeing processes. However, natural yellow dyes with good performance are rare. Traditional natural yellow dyes have issues, such as a narrow color range and poor light fastness. In this paper, a new natural yellow dye was extracted from a low-cost herb *Geranium caespitosum* L. (*G. caespitosum*). In addition, the dye composition was analyzed using UV-visible spectroscopy. The dyeing process of *G. caespitosum* dye on wool fabrics was optimized using single-factor experiments. Standard fastness tests were conducted to evaluate the sunlight, washing, and rubbing fastness of the dyed fabrics. The color characteristics and color gamut range of the dyed fabrics were evaluated. The obtained colors were compared and assessed with Pantone Matching Systems and Chinese traditional colors. The results showed that phenolic acids and flavonoids were present in the *G. caespitosum* dye solution as yellow dye compounds. The best dyeing process for wool fabrics was meta-mordanting. The dyed fabrics were bright yellow at 60 °C and golden yellow at 90 °C. All the obtained colors were in the yellow-red range. According to the ISO color fastness standards, wool fabrics dyed with *G. caespitosum* dye had good color fastness ratings, particularly sunlight fastness. Therefore, *G. caespitosum* dye is a promising natural yellow dye that can be used for wool fabric dyeing. It has advantages over other known natural yellow dyes and has certain application value.

**Keywords:** wild geranium; *Geranium caespitosum* L.; natural dyes; wool fabric; mordanting; yellow color; color fastness

## 1. Introduction

The ecological and environmental issues caused by industrial development have always been a concern. It has been estimated that over 700 newly identified pollutants, such as waste from petrochemicals, personal care products, textiles, and pesticides, have been confirmed in aquatic ecosystems in the European region. Among them, the textile and dyeing industry is considered a major source of water pollution [1]. Synthetic dyes, in particular, are widely used in various sectors, such as textiles, food, cosmetics, and pharmaceuticals [2,3]. In the textile industry alone, about 1.3 million tons of synthetic dyes are used for textile printing and dyeing annually worldwide, with approximately 8%–20% of unused dyes and auxiliary chemicals discharged into wastewater [4,5]. Dye wastewater contains high levels of biochemical oxygen demand (BOD) and chemical oxygen demand (COD) [6,7]. When dye wastewater is discharged into freshwater ecosystems, it affects the quality parameters of freshwater, such as the pH, turbidity, color, and chemical oxygen demand, etc. Discharging contaminated wastewater into the environment without any treatment poses various environmental threats, including the inhibition of photosynthesis and the death of aquatic plants [8]. As restrictions on the use of synthetic dyes gradually

increase [9], and with the widespread adoption of the green and sustainable development concept, more researchers are focusing on the study of natural dyes.

Natural dyes are dyes extracted from certain parts of plants, animals, minerals, and microorganisms. They are commonly found in plant flowers, fruits, leaves, roots, and stems, as well as in animal secretions and bacterial strains [10]. Natural dyes extracted from these sources have several advantages: they are biodegradable [11], renewable [12], and environmentally compatible, meaning they do not disrupt ecosystems. Carminic acid, a natural animal red dye, is a quinone pigment obtained from the cochineal insect, which can dye wool fabric a purplish-red color [13]. The lac resin, secreted by the lac insect, is also an animal dye that can dye fabrics purple or red [14]. Prodigiosin, extracted from the bacterium *Serratia marcescens*, is a microbial dye that can be used for textile dyeing [15]. *Fusarium oxysporum*, a fungus, can produce pink-purple anthraquinone pigments, which are also microbial dyes used for dyeing wool fabrics, giving them vivid colors [16]. Among them, plant dyes have received significant attention due to their advantages, such as being healthy and safe, biodegradable, environmentally compatible, and their therapeutic functionalities, which meet the requirements of developing eco-friendly textiles and sustainable concepts [4,17]. In addition, natural dyes can provide more natural and harmonic colors [18]. Especially, the use of metal salt mordant can change the color characteristics of dyed fabrics, greatly enriching the color gamut range of natural dyes [19] with a unique esthetic value [20]. Therefore, natural dyes have great market potential in many fields, such as textile dyeing, food, pharmaceuticals, cosmetics, etc. [21]. There is a need to explore new sources of natural dyes and their industrialization. Currently, due to issues such as high costs, insufficient raw material supply, and inadequate development of new processes, the industrial application of natural dyes has been limited [22].The abundant and sustainable resources of traditional Chinese medicinal materials ensures the positive-functioning of the Chinese medicine industry chain. However, since 2010, the production of traditional Chinese medicine in China has consistently exceeded market demand, resulting in overcapacity [23,24]. To address this issue, surplus traditional Chinese medicines can be transformed into natural dyes, allowing for the development of high-value clothing products and greater profitability. Therefore, the utilization of low-cost traditional Chinese medicinal materials for developing clothing colors holds significant economic value and has a positive impact on the prospects of applying natural dyes in industrial settings.

Natural yellow dyes from plants can be classified into different chemical types, including flavonoids, carotenoids, alkaloids, anthraquinones, diarylheptanoids, and tannins (as shown in Table 1). These dyes can be used directly or can be applied together with mordant. Most of the yellow dyes contain phenolic groups, which can form coordination complexes with metal ions, such as $Al^{3+}$ and $Cu^{2+}$, and thus, improve the color properties and fastness of dyed fabrics. Color fastness is an important indicator of dyeing quality in textiles. It refers to the ability of dyed textiles to withstand various factors and to maintain their original color during wear or subsequent processing. The higher the color fastness, the better the dyeing quality, whereas lower color fastness indicates poorer dyeing quality [25]. However, each type of yellow dye has its defects. For example, tannin-based yellow dyes have excellent color fastness properties but lower color purity. Flavonoid and alkaloid dyed fabrics have higher color purity, but poorer light fastness [26,27]. The dyeing purity or intensity of a color refers to the chroma or saturation of a color, which is an important indicator for evaluating color. It can be measured using a colorimeter, which is represented by the parameter *C\**, with a range from 0 to 100. A high *C\** value indicates a vivid, pure, and vibrant color, while a low *C\** value results in a more subdued or desaturated color appearance [28]. Dyes with similar physical and chemical properties may show similar dyeing properties, color characteristics and dyeing fastness. When applied together, some dyes may provide a synergistic effect, and improve the overall dyeing effect of the fabrics.

**Table 1.** Main chemical components of natural yellow dyes and representative plants.

| Chemical Type | Yellow Dyes | Plants |
|---|---|---|
| Flavonoids [29] | Quercetin, hyperoside, rutin, Luteolin, apigenin, etc. | *Flos Sophorae Immaturus (Sophora japonica), Reseda odorata* |
| Carotenoids [30] | Crocin, croconic acid, lutein, etc. | *Gardenia jasminoides, Tagetes erecta* |
| Alkaloids [31] | Berberine, palmatine, jatrorrhizine, etc. | *Phellodendron amurense, Coptis chinensis* |
| Anthraquinones [32] | Emodin, chrysophanol, etc. | *Rheum palmatum, Senna obtusifolia* |
| Diarylheptanoids [33] | Curcumin, etc. | *Curcuma longa* |
| Phenolic acids [34] | Gallic acid, ellagic acid, etc. | *Galla chinensis, Punica granatum* |

Wild geranium, *Geranium caespitosum* L. (*G. caespitosum*) is an annual herbaceous plant belonging to the Geranium genus of the Geraniaceae family. It is native to America and is distributed in many countries, such as China, the United States and France, etc. There are more than 60 species in the Geranium genus in China, with abundant resources and a wide use as a traditional Chinese medicine [35]. According to the literatures, the main chemical components in *G. caespitosum* extract include tannins (such as geraniin $C_{41}H_{28}O_{27}$, corilagin $C_{27}H_{22}O_{18}$), flavonoids (such as kaempferol $C_{15}H_{10}O_6$, quercetin $C_{15}H_{10}O_7$, hyperoside $C_{21}H_{20}O_{12}$), organic acids (such as gallic acid $C_7H_6O_5$, ellagic acid $C_{14}H_6O_8$), and volatile oils, etc. [35,36]. Many of these bioactive ingredients have antioxidant, anti-inflammatory, anti-viral, and anti-bacterial effects [35,37]. Among them, many are natural yellow dye compounds [38–40], which can be used for textile dyeing, such as ellagic acid, gallic acid, hyperoside, quercetin, etc., as shown in Table 1. The molecular structures of ellagic acid, gallic acid, hyperoside, and quercetin, are shown in Figure 1.

**Figure 1.** Chemical structures of the dye molecules in *G. caespitosum*. (**a**) Ellagic acid. (**b**) Gallic acid. (**c**) Hyperoside. (**d**) Quercetin.

Natural dyes exhibit a high affinity for natural fibers, particularly for protein-based fibers. Wool, belonging to the protein fiber category, primarily consists of keratin, which consists of long chains of amino acids linked together by peptide bonds. Within the keratin structure, various types of chemical bonds and intermolecular forces play a role in determining the properties of wool. These include disulfide bonds, ionic bonds, hydrogen bonds, and van der Waals forces. The combination of these bonding forces and interactions results in a three-dimensional network of intertwined keratin chains, forming the complex and unique structure of wool. This advanced structure provides wool with its desirable properties, such as elasticity, resilience, moisture absorption, insulation, and durability. Wool is recognized as a high-quality and mid-to-high-end fabric [41]. The molecular structure of wool contains a large number of alkaline and acidic side chain groups, exhibiting amphoteric properties in aqueous solutions [42,43]. Under alkaline conditions, the protein's polypeptide chains break, disulfide bonds hydrolyze, and ionic bonds dissociate, leading to damage to the keratin structure, resulting in yellowing of the wool, reduced sulfur content, and partial dissolution [42,43]. Under acidic conditions, the ionic bonds in the keratin structure dissociate; furthermore, wool fabrics have a high affinity for anionic plant dyes [44].

Currently, there are many studies on the use of *G. caespitosum* extracts in the medical field [35]. However, there are few reports on its dyeing properties in the textile domain. In this study, a hot water extract of *G. caespitosum* has been analyzed using UV-visible spectroscopy. Single-factor experiments were conducted to optimize the parameters of the dyeing process. The color characteristic values and the color strength (*K/S*) values were used as evaluation criteria to assess the dyeing properties, color gamut range, color values, and color fastness of the dyed wool fabrics. The interaction between the dye molecules, metal mordants, and wool fabrics were further discussed.

## 2. Experiments

### 2.1. Materials and Chemicals

The plain weave wool fabric used in the experiment (purity 100%, fabric weight 208 g/m$^2$, warp yarn density 28 tex, weft yarn density 24 tex) was purchased from the local market as a bleached product. The dried *G. caespitosum* whole plant was purchased from Beijing Huaishuntang Pharmaceutical Co., Ltd., Beijing, China, and its origin was Bozhou, China. Citric acid ($C_6H_8O_7$) was purchased from Tianjin Tianli Chemical Reagent Co., Ltd., Tianjin, China. Sodium hydroxide (NaOH) was provided by Tianjin Hengxing Chemical Reagent Co., Ltd., Tianjin, China. In addition, Tin(II) chloride dihydrate ($SnCl_2 \cdot 2H_2O$) was purchased from Tianjin Beichen Fangzheng Reagent Factory, Tianjin, China. All the above chemicals were of analytical grade. Distilled water was made in the laboratory.

### 2.2. Wool Fabrics Pre-Treatment

Before dyeing, the wool fabric samples were immersed in a water solution containing 5 g/L of synthetic detergent (non-ionic surfactant) and washed at 50 °C for 30 min. Subsequently, they were washed with distilled water and dried at room temperature [45].

### 2.3. G. caespitosum Dye Extraction

The dry plants of *G. caespitosum* were washed with distilled water and dried at 60 °C in the oven to a constant weight. Then, they were ground for 0.5 min using a grinder at a speed of 24,000 r/min, and the residue was further ground for 0.5 min after passing it through a 40-mesh sieve to obtain a uniform powder. The powder was extracted with distilled water at a weight/volume ratio of 1:10. The extraction was carried out under constant reflux at 100 °C for 1.5 h, and the *G. caespitosum* dye extract was obtained after filtration. The dye extract was then freeze-dried to obtain a *G. caespitosum* dye powder. The concentration of the *G. caespitosum* dye extract was calculated as 11 g/L, which was used for the dyeing process.

### 2.4. Dyeing Process

In this study, a *G. caespitosum* dyeing process on wool fabrics was carried out without and with mordant (pre-mordanting, meta-mordanting, and post-mordanting). In the dyeing process without mordant, the wool fabrics were directly immersed in the dye solution. In the pre-mordanting method, the wool fabrics were first immersed in the mordant solution and then dyed. In the meta-mordanting method, the wool fabrics were dyed in a dye solution containing the mordant. In the post-mordanting method, the dyed wool fabrics were then immersed in the mordant solution. All experiments were carried out with a liquor ratio of 1:50 and used citric acid and sodium hydroxide as pH buffering agents. The dye solution was prepared as a dye bath, and the temperature was raised to a set temperature at a rate of 1.5 °C/min starting at 30 °C. After the dyeing process, the fabrics were thoroughly washed with distilled water and dried in the shade at room temperature. The mordanting temperature was 60 °C, and the mordanting time was 15 min.

To optimize the dyeing process, the dyeing temperature, dyeing time, dye bath pH value, mordant concentration, and dye concentration were set as the influencing factors, respectively. Color characteristic values were used as the evaluation indicators to conduct

the single-factor optimization experiments. Table 2 lists the parameters in the experiments. Figure 2 shows the dyeing process.

**Table 2.** Optimization of the dyeing process using single-factor experiments.

| Influence Factor | Fixed Factors | | | | | Levels |
|---|---|---|---|---|---|---|
| | Dyeing Temperature (°C) | Dyeing Time (min) | pH | Mordant Concentration (g/L) | Dye Concentration (g/L) | |
| Dyeing temperature (°C) | — | 90 | 4.0 | 0 | 11 | 30, 40, 50, 60, 70, 80, 90, 100 |
| Dyeing time (min) | 60, 90 | — | 4.0 | 0 | 11 | 10, 20, 30, 40, 50, 60, 70, 80, 90, 100 |
| Dye bath PH | 60, 90 | 90 | — | 0 | 11 | 4, 5, 6, 7, 8, 9, 10, 11 |
| Mordant concentration (g/L) | 60, 90 | 90 | 4.0 | — | 11 | 0.01, 0.02, 0.04, 0.08, 0.16, 0.31, 0.63, 1.25, 2.50, 5.00, 10.00, 20.00 |
| Dye concentration (g/L) | 60, 90 | 90 | 4.0 | 10 | — | 0.01, 0.02, 0.04, 0.09, 0.17, 0.34, 0.69, 1.38, 2.75, 5.50, 11.00, 22.00, 33.00 |

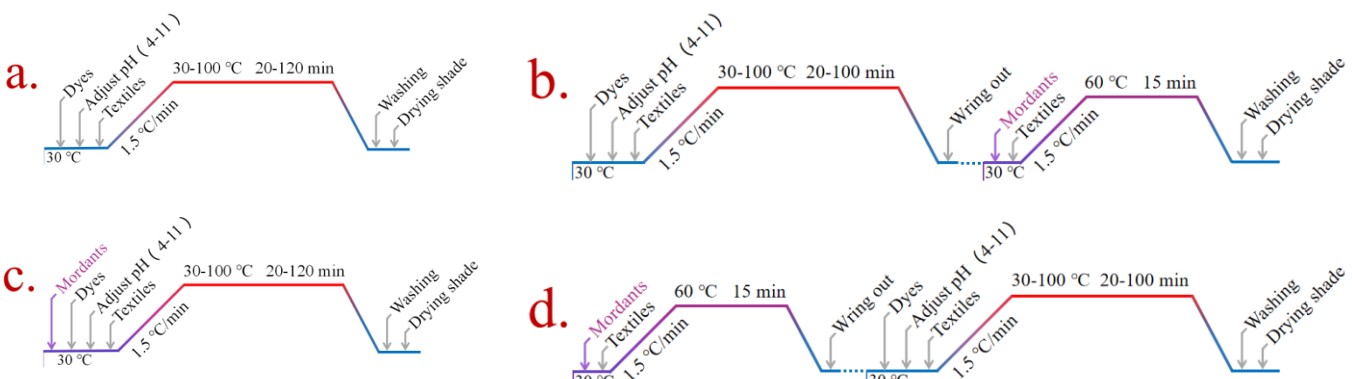

**Figure 2.** Scheme of dyeing process: (**a**) dyeing without mordant; (**b**) post–mordanting; (**c**) meta–mordanting; (**d**) pre–mordanting.

*2.5. UV-Visible Spectroscopy*

The *G. caespitosum* dye solution was diluted 100 times and scanned in the wavelength range of 200–800 nm using a Lambda 35 UV-Vis spectrophotometer (New York, PE, USA).

*2.6. Color Characteristics and K/S Value Measurements*

The dyed fabrics were measured for the *L\**, *a\**, *b\**, *C\**, *h°*, *ΔE*, and *K/S* values using a desktop spectrophotometer (Color Eye® 7000A, X-rite, USA) under a D65 light source and a 10° observation angle. *L\** represents lightness (0 (black)–100 (white)), with higher values indicating brighter colors. *a\** represents values on the red-green axis (−128 (green)–0–127 (magenta)), with positive values (0–127) indicating red colors (larger values indicating a deeper red color), and negative values (−128–0) indicating green colors (smaller values indicating a greener color). *b\** represents values on the yellow-blue axis (−128 (blue)–0–127 (yellow)), with positive values (0–127) indicating yellow colors (higher values indicating a deeper yellow color), and negative values (−128–0) indicating blue colors (smaller values indicating a bluer color). *C\** represents purity (ranging from 0 to 100), with higher values indicating purer colors. The *C\** value is calculated according to Formula (1). *h°* represents the hue angle (0° (red)–60° (yellow)–120° (green)–180° (cyan)–240° (blue)–300° (magenta)–360° (red)), which is mainly used to distinguish different colors. The *h°* value is calculated according to Formula (2). *ΔE* value represents the overall color difference of the dyed sample, with larger values indicating a greater difference from the color of the undyed fabric, and smaller values indicating a smaller difference. The *ΔE* value is calculated according to Formula (3). The *K/S* value represents color strength, with higher values indicating deeper colors and lower values indicating lighter colors. The *K/S* value is calculated according to the Kubelka–Munk Equation (4). To ensure the accuracy of the

experiment, the dyed fabric was folded into four layers; then, three random test points were measured and the average value was taken.

$$(C^*) = \sqrt{a^2 + b^2} \tag{1}$$

$$(h^\circ) = \tan^{-1}\left(\frac{b}{a}\right) \tag{2}$$

$$\Delta E = \sqrt{(\Delta L^*)^2 + (\Delta a^*)^2 + (\Delta b^*)^2} \tag{3}$$

$\Delta L^*$, $\Delta a^*$ and $\Delta b^*$ represent, respectively, the difference of $L^*$, $a^*$ and $b^*$ value of the dyed sample and undyed sample.

$$K/S = \frac{(1-R)^2}{2R} - \frac{(1-R_0)^2}{2R_0} \tag{4}$$

$R_0$ is the remission fraction from the undyed sample, $R$ is the remission fraction from the dyed sample (under $\lambda_{\max}$), $K$ and $S$ are the absorption and scattering coefficients of the dyed sample.

### 2.7. Color Fastness Measurements

Color fastness experiments of all the wool fabric samples were carried out according to the ISO standards. Washing fastness and rubbing fastness tests were carried out according to ISO 105-C10: 2006 and ISO 105-X12: 2001. Grey scale grades (5-Excellent, 3-Fair, 1-Very Poor) evaluate the results of washing fastness from 1 to 5.

Light fastness was carried out according to ISO 105-B02: 1994/Amd. 2:2000. Blue scale grades evaluate the light fastness property with values from 1 to 8 (8-Excellent, 4-Fair, 1-Very Poor).

## 3. Results and Discussion

### 3.1. UV-Visible Spectroscopy of G. caespitosum Extract

Plant dyes typically contain chromophores in their structure. Chromophores must contain a conjugate system that can absorb visible electromagnetic radiation in the UV-visible range, thus, causing electron transition [46]. In order to qualitatively analyze the presence and types of these plant secondary metabolites in the *G. caespitosum* extract, we used UV-visible spectroscopy. Figure 3 shows that the *G. caespitosum* extract has obvious absorption in the ultraviolet range (200–400 nm), with strong absorption peaks at 216 nm and 268 nm, and a weaker absorption peak around 360 nm. There was no absorption in the visible light range (400–800 nm). In comparison with the literature, it can be inferred that the *G. caespitosum* extract contains phenolic acid and flavonoid components, with gallic acid at 216 nm [47] and 268 nm [48], ellagic acid at 357 nm [49], hyperoside at 363 nm [50], and quercetin at 374 nm [51]. However, more precise analysis is currently unavailable and further studies will be carried out in the future.

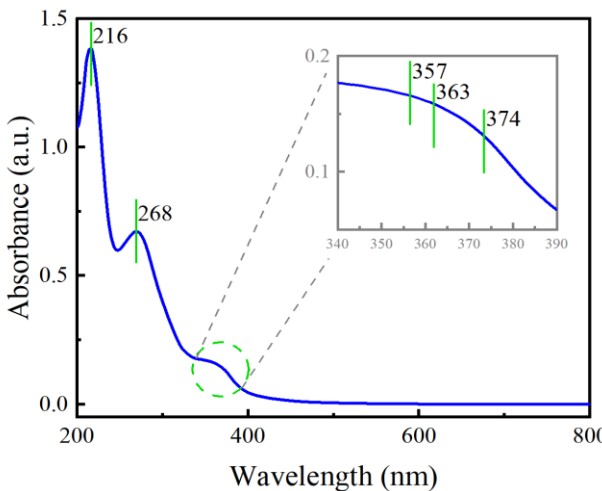

**Figure 3.** UV-Visible spectrum of the *G. caespitosum* extract solution.

### 3.2. *Optimization of the G. caespitosum Dyeing Process on Wool Fabrics*

Parameters of the *G. caespitosum* dyeing process (without mordant) on wool fabrics were optimized by changing the dyeing temperature, dyeing time, and the dye bath's pH.

### 3.2.1. Optimization of Dyeing Temperature

Dyeing temperature is an important factor that affects the dyeing quality, as it affects the kinetic energy of dye molecules in the dye bath and the degree of expansion of the wool fibers. Figure 4a shows the color characteristic values of dyed fabrics under different dyeing temperatures. As the dyeing temperature increased, the $L^*$ value gradually decreased and then stabilized, while the $a^*$ and $b^*$ values first increased, slowly, and then decreased before stabilizing. At 60 °C, the $a^*$ and $b^*$ values both reached their highest points ($a^*$: 11.58, $b^*$: 36.24), with yellow as the main color and red as the secondary color. In addition, the $C^*$ value reached the highest point (38.04). Furthermore, the $h°$ value was 72.27°. The apparent color of the dyed wool fabrics was brownish-yellow. When the temperature increased from 60 °C to 90 °C, the color characteristic values decreased and reached a stable state, with the $L^*$ value from 50.72 to 32.24, the $a^*$ value from 11.58 to 10.78, the $b^*$ value from 36.24 to 24.98, the $C^*$ value from 38.04 to 27.20, and the $h°$ value shifting from 72.27° to 66.66°. In addition, the apparent color of the dyed wool fabrics changed from brown-yellow at 60 °C to dark brown at 90 °C, which were in the yellow-red region.

The scales on the surface of wool fibers are composed of dense keratin cells. When the temperature is low, the scale structure cannot completely dissolve and open up, making it difficult for dye molecules (such as gallic acid of 170.12 Da, ellagic acid of 302.28 Da, hyperoside of 464.38 Da, and quercetin of 302.24 Da) to diffuse into the interior of the fibers, resulting in a lower adsorption amount and lighter colored wool fabrics. When the temperature is high, the scale-shaped structure completely dissolves and opens up. The kinetic energy of the dye molecules increases and the diffusion rate is higher, making it easier for the dye molecules to diffuse into the interior of the fibers. However, excessively high temperatures can weaken the hydrogen bonds and van der Waals forces between the wool fabrics and dye molecules, leading to desorption. Considering the dyeing properties and color differences of the dyed wool fabrics, both dyeing temperatures, 60 °C and 90 °C, were selected for further studies.

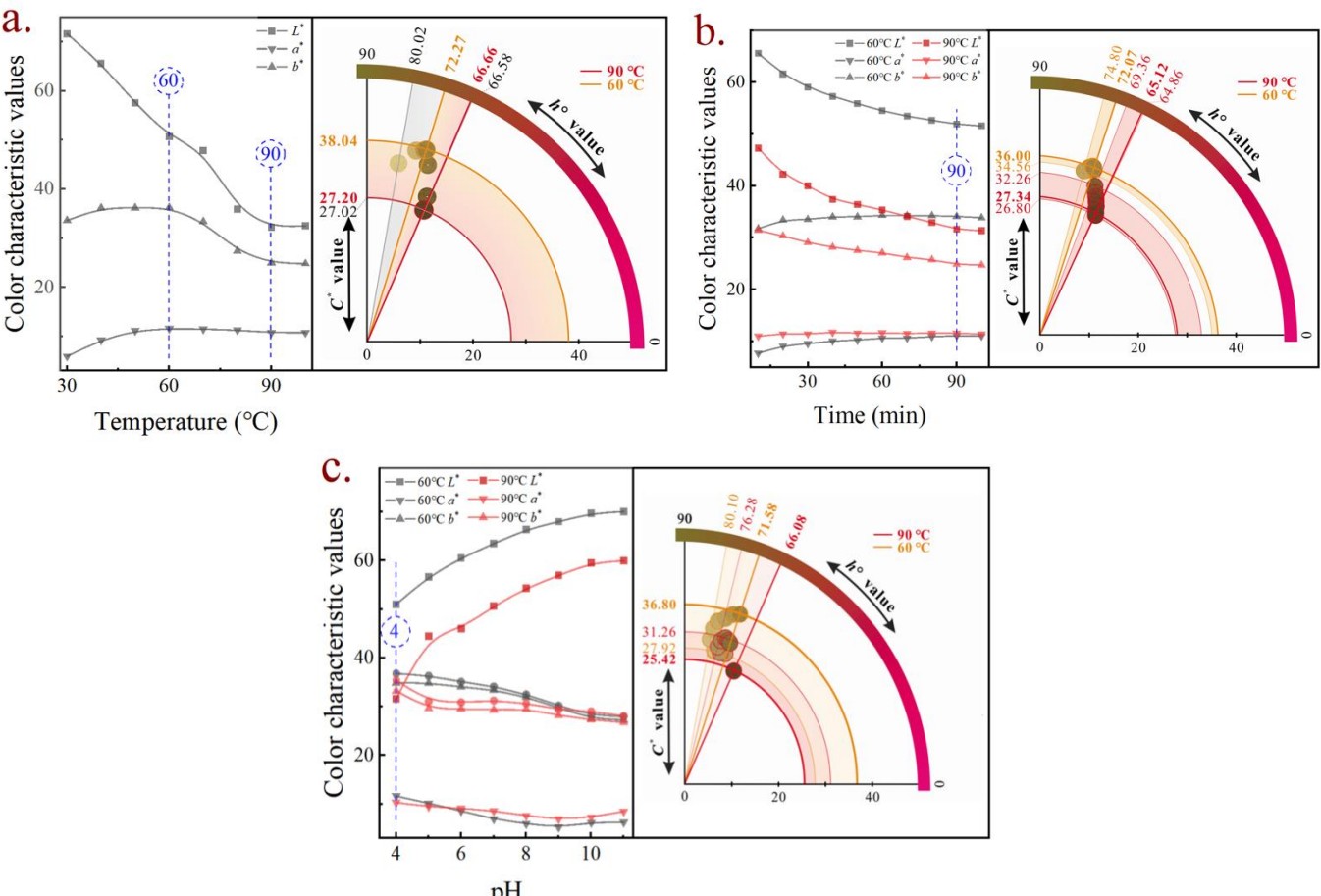

**Figure 4.** Effect of the dyeing parameters on the color characteristic values of *G. caespitosum* dyed wool fabrics: (**a**) dyeing temperature; (**b**) dyeing time; (**c**) dye bath pH value.

### 3.2.2. Optimization of Dyeing Time

Dyeing time is an important factor affecting a dye's uptake, dyeing uniformity and color fastness. In actual production, the dyeing time can be appropriately shortened according to the properties of the dye and the requirements of the dyeing color, in order to increase dyeing productivity. Figure 4b shows the color characteristic values of dyed fabrics at different dyeing times. As the dyeing time increased, the $L^*$ value gradually decreased, and the $b^*$ value slowly increased. The $a^*$ value gradually decreased at 90 °C, while it increased slowly at 60 °C. When the dyeing time reached 90 min, all color characteristic values stabilized. The $C^*$ value reached its maximum 36.00 at 60 °C and 27.34 at 90 °C. The $h°$ value was 72.07° at 60 °C and 65.12° at 90 °C. At this time, the wool fibers have reached adsorption equilibrium. Prolonging the dyeing time will cause desorption [52] and increase the cost. Therefore, the optimal dyeing time was selected as 90 min.

### 3.2.3. Optimization of Dye Bath pH

The pH of the dye solution is one of the most important factors affecting the dyeing process. As the pH of the dye solution increased, the $L^*$ value gradually increased, and the $a^*$ and $b^*$ values gradually decreased. The apparent color of the dyed wool fabrics became gradually lighter. When the pH was 4.0, both the $a^*$ and $b^*$ values reached the highest point with an $a^*$ of 11.62 and a $b^*$ of 34.90 at 60 °C, and with an $a^*$ of 10.30 and a $b^*$ of 23.24 at 90 °C. The $L^*$ value reached its lowest value of 51.00 at 60 °C and of 31.60 at 90 °C. The $C^*$ value was at its maximum value of 36.80 at 60 °C and at its minimum value of 25.24 at 90 °C. The $h°$ value was 71.58° at 60 °C and 66.08° at 90 °C. At this point, the color characteristic values were optimal.

The *G. caespitosum* dye contains molecules with a large amount of hydroxyl and carboxylic acid groups. In acidic solutions, the dye molecules carry a negative charge. The isoelectric point of wool fibers is 4.2–4.8. Within this pH range, the wool fibers are in equilibrium and do not adsorb dyes [1]. When the pH is greater than 4.8, the wool fibers carry a negative charge, resulting in increased electrostatic repulsion between the fibers and dye molecules, which is unfavorable for dye adsorption and results in lighter dyeing. When the pH is less than 4.2, the wool fibers carry a positive charge, resulting in an increased electrostatic attraction between the fibers and the dye molecules, leading to increased dye adsorption and deeper dyeing. Under extreme acidic or alkaline conditions, the protonation of the carboxyl groups or the deprotonation of the amino groups can break the ionic bond between the carboxyl anion and the ammonium ion, resulting in the lowest level of ionic bonds in keratin and a decrease in fiber strength. Under alkaline conditions, the wool strength and other mechanical properties decrease, and the wool fibers become yellow and rough [43]. Therefore, the optimal pH value of the dye solution is 4.0.

Above all, the optimized dyeing conditions of wool fabrics dyed with *G. caespitosum* dye were a dyeing temperature of 60 °C and 90 °C, dyeing time of 90 min, and a pH of 4.0; these were further used for the following mordanting processes.

### 3.3. Effect of Mordanting on Color Characteristic Values of G. caespitosum Dyed Wool Fabrics

Using metal mordants is a basic step in natural dyeing. Metal ion mordants not only provide dyed fabrics with different apparent colors but also enhance color fastness. In this article, $SnCl_2$ was selected as the mordant, and pre-mordanting, meta-mordanting, and post-mordanting processes were evaluated at 60 °C and 90 °C. The color characteristics and *K/S* values of the wool fabrics were measured, as shown in Table 3. The results showed that $Sn^{+2}$ ions significantly changed the color of dyed wool fabrics. In addition, different mordanting processes have different effects on the hue angle $h°$. The range of the $h°$ value was between 2° and 6°. The color purity, $C^*$ value, was in the order of meta-mordanting > pre-mordanting > dyeing without mordant > post-mordanting, indicating that meta-mordanting was the best dyeing process regarding color purity. Through the meta-mordanting process, the color of the *G. caespitosum* dyed wool fabrics was bright yellow at 60 °C and golden yellow at 90 °C. In addition, the $L^*$ value was the highest of 75.14 at 60 °C and of 55.66 at 90 °C. The $b^*$ value was highest of 64.98 at 60 °C and of 69.74 at 90 °C. The $C^*$ value was highest of 66.02 at 60 °C and of 60.76 at 90 °C. The difference in color characteristic values between the meta-mordanting dyed fabrics and the dyed fabrics without mordanting was in the order of $C^*$ value > $b^*$ value > $a^*$ value > $L^*$ value, with the $b^*$ value (indicating light yellow) and $C^*$ value having the greatest increase, which is rare and typical in the dyeing properties of natural yellow dyes.

This phenomenon can be explained by the chelation mechanism of the interaction between the dye molecules, mordant, and wool fibers, as shown in Figure 5. Figure 5 illustrates the coordination bonding mode of $Sn^{+2}$ ions with dyes and wool fibers. The coordination number of $Sn^{+2}$ ions is four, forming a coordination entity with the active sites of wool fibers (i.e., the hydroxyl and amino groups). This coordination entity forms a non-soluble ternary complex, where two sites are bound to the dye molecules, and the other site is bound to the wool fiber [50,53,54]. During meta-mordanting, $Sn^{+2}$ ions form a complex with the lone pair of electrons of the carboxyl and amino groups of wool fibers, and then form a complex with the hydroxyl groups of the dye molecules through the 3P vacant orbital. This enhances both color intensity and dye fastness. Wool fabrics dyed with *G. caespitosum* dye through meta-mordanting with Tin(II) chloride has the best color performance. The color values and their applications will be further explored.

**Table 3.** Effect of mordanting on the color characteristic values and the *K/S* values of *G. caespitosum* dyed wool fabrics.

| Mordanting | Dyeing Temperature | Color Characteristic Values | | | | | K/S Value ($\lambda_{max}$ = 390 nm) | Sample |
|---|---|---|---|---|---|---|---|---|
| | | L* | a* | b* | C* | h° | | |
| Without mordant | 60 | 52.68 | 13.22 | 38.26 | 48.12 | 74.06 | 25.05 | |
| Pre-mordanting | 60 | 63.94 | 13.86 | 52.30 | 54.12 | 75.16 | 19.67 | |
| Meta-mordanting | 60 | 75.14 | 11.72 | 64.98 | 66.02 | 79.78 | 14.95 | |
| Post-mordanting | 60 | 56.70 | 11.00 | 34.32 | 36.04 | 72.22 | 22.16 | |
| Without mordant | 90 | 32.44 | 14.16 | 30.48 | 33.60 | 65.08 | 23.64 | |
| Pre-mordanting | 90 | 52.68 | 19.02 | 55.92 | 59.06 | 71.20 | 22.93 | |
| Meta-mordanting | 90 | 55.66 | 29.08 | 69.74 | 60.76 | 67.38 | 25.05 | |
| Post-mordanting | 90 | 34.14 | 11.92 | 26.60 | 29.12 | 65.82 | 28.25 | |

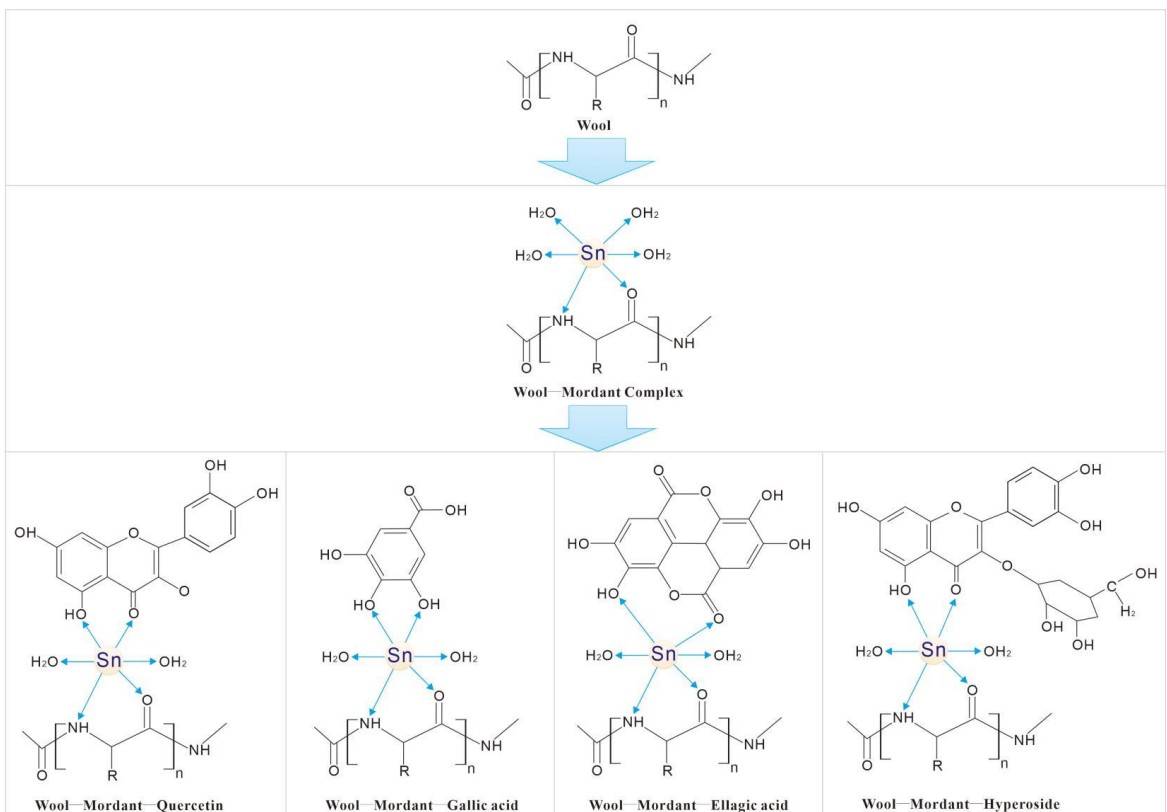

**Figure 5.** Schematic representation of the interaction between the wool fibers, mordant and dye molecules.

Using the meta-mordanting process, the effect of the concentration of the mordant on the color characteristics of the dyed fabrics is shown in Figure 6. As the mordant concentration increased, the *L\** value gradually increased, indicating a decrease in dye absorption. This suggests that the dye was desorbed from the fabrics and interacted with the mordant to form insoluble complexes. The *a\** and *b\** values increased significantly. When the mordant concentration reached 10 g/L, except for the continuing increase in the *b\** value at 60 °C, most of the other color characteristic values reached their maximum

point. This indicated that the binding sites (carboxyl and amino groups) on the fabrics were exhausted and had reached a saturation state. In addition, the color of the dyed fabrics gradually changed from brownish-yellow (dyeing without mordant) to the optimal dyeing color. The hue angle $h°$ shifted noticeably. When the concentration reached 10 g/L, the $L*$ value (72.84 at 60 °C and 54.30 at 90 °C), $a*$ value (13.92 at 60 °C and 23.40 at 90 °C), $b*$ value (64.78 at 60 °C and 63.90 at 90 °C), and $C*$ value (66.24 at 60 °C and 68.04 at 90 °C) all stabilized. The $h°$ value was 77.84° at 60 °C and the color was bright yellow. At 90 °C, the $h°$ value was 69.90° and the color was golden. Therefore, the optimal mordant concentration was 10 g/L.

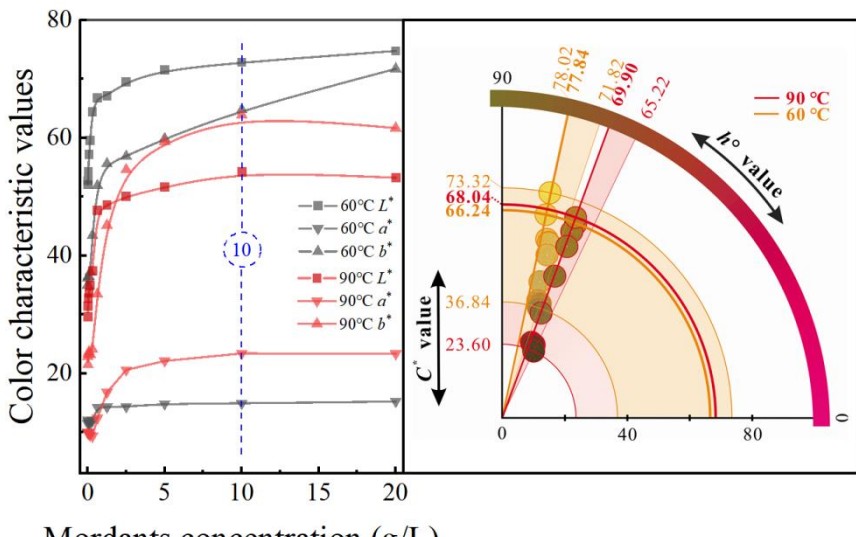

**Figure 6.** Effect of the mordant concentration on the color characteristic values of *G. caespitosum* dyed wool fabrics.

### 3.4. Color Characteristics and Color Evaluation of G. caespitosum Dyed Wool Fabrics

In the actual dyeing process, the dye concentration can control the color characteristics of the dyed fabrics, which plays an important role in determining the color of dyed fabrics and the production cost. The results of the color characteristic values and the color evaluation of *G. caespitosum* dyed wool fabrics at different dye concentration are shown in Table 4 and Figure 7.

**Table 4.** Effect of dye concentration on the color characteristic values and color evaluation of *G. caespitosum* dyed wool fabrics.

| Temperature (°C) | Dye Concentration (g/L) | Color Characteristic Values and Color Difference Value | | | | | | PANTONE (TCX) | | | Traditional Chinese Colors | |
|---|---|---|---|---|---|---|---|---|---|---|---|---|
| | | $L*$ | $a*$ | $b*$ | $C*$ | $h°$ | $\Delta E$ | Number | Name | $\Delta E$ | Name | $\Delta E$ |
| 90 | 22.00 | 48.86 | 25.94 | 58.58 | 64.07 | 66.12 | 6.73 | — | — | — | Agate | 2.40 |
| 90 | 11.00 | 54.92 | 26.30 | 64.90 | 70.00 | 67.90 | 7.22 | — | — | — | Barn yellow | 1.99 |
| 90 | 5.50 | 61.66 | 30.64 | 78.70 | 84.44 | 68.70 | 5.23 | — | — | — | Sauce orange | 2.90 |
| 90 | 2.75 | 66.48 | 26.74 | 81.36 | 85.62 | 71.82 | 4.86 | — | — | — | Shrimp yellow | 2.47 |
| 90 | 1.38 | 70.72 | 20.62 | 77.60 | 80.32 | 75.14 | 10.14 | 14-0760 | Cyber Yellow | 2.70 | Orange | 0.68 |
| 90 | 0.17 | 76.72 | 7.68 | 43.40 | 44.04 | 79.98 | 3.82 | 14-0827 | Dusky Citron | 3.20 | White oak | 2.18 |
| 90 | 0.09 | 78.82 | 4.96 | 34.48 | 34.84 | 81.80 | 7.09 | 14-0935 | Jojoba | 4.40 | Silk color | 2.88 |
| 90 | 0.01 | 84.14 | 1.22 | 21.86 | 21.90 | 86.80 | — | 13-0915 | Reed Yellow | 2.80 | Beige | 1.35 |
| 60 | 22.00 | 68.48 | 19.44 | 69.24 | 71.92 | 74.34 | 5.07 | 15-0955 | Old Gold | 3.50 | Sandalwood | 0.98 |
| 60 | 11.00 | 73.10 | 14.78 | 65.92 | 67.56 | 77.36 | 3.73 | 14-0852 | Freesia | 2.50 | Sumach | 0.46 |
| 60 | 5.50 | 76.14 | 10.92 | 60.82 | 61.82 | 79.84 | 2.67 | 14-0754 | Super Lemon | 2.70 | Loquat color | 1.94 |
| 60 | 2.75 | 77.72 | 8.34 | 53.44 | 54.08 | 81.14 | 6.56 | 13-0755 | Primrose Yellow | 4.00 | Maize | 2.09 |
| 60 | 0.34 | 82.14 | 2.90 | 38.62 | 38.74 | 85.64 | 5.34 | 12-0720 | Mellow Yellow | 3.00 | White oak | 2.99 |
| 60 | 0.09 | 84.28 | 0.74 | 25.66 | 25.70 | 88.40 | 4.68 | 11-0616 | Pastel Yellow | 4.40 | White tea | 2.37 |
| 60 | 0.01 | 86.30 | -0.12 | 16.64 | 16.64 | 90.46 | — | 11-0107 | Papyrus | 4.30 | Branch yellow | 0.53 |

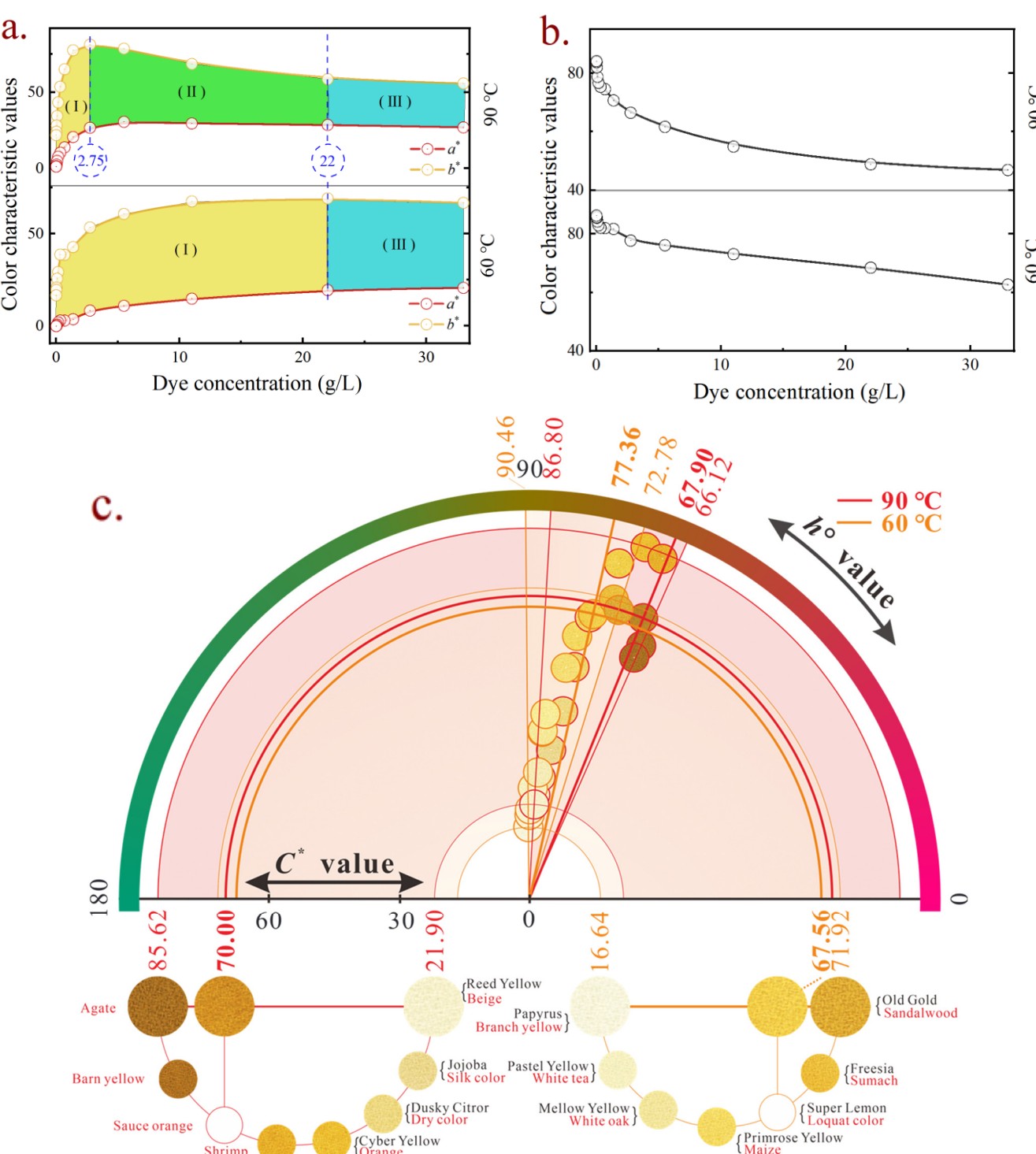

**Figure 7.** Effect of dye concentration on the color characteristic values of *G. caespitosum* dyed wool fabrics: (**a**) *a\** and *b\** values; (**b**) *L\** value; (**c**) *C\** and *h°* values.

Figure 7a shows the effect of the *G. caespitosum* dye concentration on the *a\** and *b\** values of the dyed wool fabrics. With the increase in dye concentration, the difference between the *a\** and *b\** values of the dyed wool fabrics changed in three stages. In stage I, the equidistant difference between the *a\** and *b\** values increased from small to large, with the change of *b\** value significantly larger than that of the *a\** value, mainly showing light yellow. When the dye concentration reached 2.75 g/L, the *b\** value reached the highest point (81.36). In this stage, with the increase in dye concentration, the light yellow gradually

strengthened, with the $a^*$ and $b^*$ values gradually increasing, producing different strengths of yellow colors. In stage II, at 90 °C, the equidistant difference between the $a^*$ and $b^*$ values decreased from large to small, and the $b^*$ value (yellow light) decreased, while the $a^*$ value (light red) slowly raised and stabilized. This stage indicated that excessive dye concentration would lead to a decrease in the $b^*$ value and $C^*$ value. In stage III, the equidistant difference between the $a^*$ and $b^*$ values was similar, and the color change tended to be stable. Continuing to increase the dye concentration had little effect on the $a^*$ and $b^*$ values. When the dye concentration reached 22 g/L, the color of the dyed wool fabrics was stabilized at both 90 °C and 60 °C.

From Figure 7b,c, as the dye concentration increased, the $L^*$ value gradually decreased as the equidistant difference of the $a^*$ and $b^*$ values changed from stage I to stage III. The range of the $C^*$ value gradually increased, and the $h°$ value produced a shift of 17°–20°. When the dye concentration reached 22 g/L, the $C^*$ values were 71.92 and 64.07 at 60 °C and 90 °C, respectively, and the $h°$ values were 74.34° and 66.12°, respectively. This is because increasing the dye concentration can increase the coloring strength and adsorption of wool fibers. However, the amount of dye molecules bound to wool fibers is limited. When the amount of dye absorbed or diffused reaches saturation, the color characteristic values tend to be stable [55]. Therefore, the optimal dye concentration at 60 °C and 90 °C was 22 g/L.

In order to evaluate the obtained colors, the dye concentration was set as the variable and the color difference value $\Delta E \geq 3$ was used as the evaluation standard, according to the *Names and colorimetric characteristics of traditional colors in China* (GB/T 31430-2015). The number of color samples that showed significant differences was counted and their color gamut range at 90 °C and 60 °C was listed. Furthermore, these samples were compared with the internationally recognized Pantone Matching System to evaluate their value for practical applications. As in Table 4, out of fifteen color samples, there were five colors that matched the Pantone colors ($\Delta E \leq 3$), six colors that were similar ($3 < \Delta E \leq 5$), and four colors that were different ($\Delta E > 5$).

As shown in Table 4, under the conditions of a dyeing temperature at 60 °C and a dye concentration at 5.50 g/L, the color characteristic values of the dyed samples were $L^* = 76.14$, $a^* = 10.92$, $b^* = 60.82$, $C^* = 61.82$, and $h° = 79.84$, which was a bright yellow color. Under the conditions of a dyeing temperature at 90 °C and a dye concentration at 0.17 g/L, the color characteristic values of the dyed samples were $L^* = 76.72$, $a^* = 7.68$, $b^* = 43.40$, $C^* = 44.04$, $h° = 79.98$, which was golden yellow. Comparing these with traditional Chinese colors, both colors are relatively precious in traditional Chinese clothing [56].

Above all, some of the colors of wool fabrics dyed with *G. caespitosum* dye were similar to the Pantone colors, and most colors were similar to traditional Chinese clothing colors. The obtained colors had both international and ethnic characteristics and reflected their values.

### 3.5. Color Fastness

Metal mordants can usually form complexes with fibers and dyes, improving their dyeing performance and color fastness [57]. Table 5 shows the fastness of *G. caespitosum* dyed wool fabrics with and without mordanting. It can be seen that the mordanting process can significantly improve the light fastness of dyed fabrics, which increased from 3–4 to 4–5. There was no significant difference in rubbing and soap washing fastness. This indicates that $Sn^{+2}$ ions have formed strong coordination bonds between the *G. caespitosum* dye molecules and the wool fibers, enhancing the binding force and, thereby, improving the light fastness of the dyed fabrics. Compared with traditional natural yellow dyes, such as *Phellodendron amurense*, *Flos Sophorae Immaturus (Sophora japonica)*, and *Gardenia jasminoides* [27,58,59], the light fastness of *G. caespitosum* dyed fabrics was slightly higher, which made it a better natural yellow dye.

**Table 5.** Color fastness of *G. caespitosum* dyed wool fabrics.

| Wool Fabrics. | Light Fastness | Rubbing Fastness | | Washing Fastness | |
|:---:|:---:|:---:|:---:|:---:|:---:|
| | | Wet | Dry | Change | Staining |
| 60 °C without mordant | 3–4 | 3–4 | 4 | 3 | 3–4 |
| 90 °C without mordant | 3–4 | 4 | 4 | 3 | 4 |
| 60 °C $Sn^{2+}$ meta-mordanting | 4–5 | 4 | 4 | 3 | 4 |
| 90 °C $Sn^{2+}$ meta-mordanting | 4–5 | 4 | 4 | 3 | 4 |

## 4. Conclusions

Expanding the color range of natural dyes is helpful in promoting the application of natural dyes. In this study, we selected *G. caespitosum*, a low-cost traditional Chinese medicinal material, as a natural dye for wool fabric dyeing, and explored its potential as a sustainable natural yellow dye. Through simple extraction and dyeing techniques, we successfully obtained *G. caespitosum* dyed wool fabrics with rich yellow-red tones, especially bright yellow and golden yellow, which corresponded to the characteristics of traditional Chinese clothing colors [60]. Through standard fastness tests, we found that *G. caespitosum* dyed wool fabrics had excellent fastness performance, which can meet the daily wear standards. Compared with other traditional natural yellow dyes, *G. caespitosum* dye has advantages, such as stable and rich raw material sources, and low prices. It has great development potential and can be used as an ecological, efficient, and sustainable natural yellow dye. In the field of textiles, *G. caespitosum* dye offers high added value on wool fabrics and conforms to the criteria of the application of nature dyes.

**Author Contributions:** Conceptualization, Z.Z. and C.Y.; methodology, Z.Z.; software, C.Y.; validation, C.Y. and Z.Z.; formal analysis, Z.Z., J.L. and C.Y.; investigation, Z.Z.; resources, F.X.; data curation, C.Y. and Z.Z. writing—original draft preparation, C.Y. and Z.Z.; writing—review and editing, Z.Z.; visualization, C.Y.; supervision, Z.Z.; project administration, Z.Z.; funding acquisition, F.X. All authors have read and agreed to the published version of the manuscript.

**Funding:** This work was supported by the key research topics of economic and social development in Heilongjiang Province (grant number 22330) and Basic Scientifc Research Expenses for Heilongjiang Provincial Colleges and Universities Research on Project (grant number 145209607).

**Institutional Review Board Statement:** Not applicable.

**Informed Consent Statement:** Not applicable.

**Data Availability Statement:** The data presented in this work are available in the article.

**Conflicts of Interest:** The authors declare no conflict of interest.

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
