# Peer review of "Study on Dyeing Properties and Color Characteristics of Wool Fabrics Dyed with Geranium caespitosum L. Extract—A New Natural Yellow Dye"

_coatings, doi:10.3390/coatings13061125_

Round 1

Reviewer 1 Report

Authors mentioned, "Tannins have good dyeing fastness properties better but lower dyeing color purity, while flavonoids (and alkaloids) have higher dyeing color purity but poor light-fastness properties." 

Comment 1. What do authors mean by dyeing color purity? Are they indicating the labeling effect of dyeing OR the purity of dyes (in general)? Please state clearly.

Continue to the previous text mentioned by the authors in the manuscript.

"Dyes with similar physical and chemical properties show similar dyeing properties, color characteristics, and dyeing fastness"

Comment 2. This may be true but limited to only some cases, as substitution or modification of any particular position on chromophores or natural pigments can drastically change their color properties. Say example, naturally-derived anthraquinones are sensitive to any small modification on the ring, even so the position of substitution in the ring, and therefore any subtle change can alter the color and fastness properties. 

Continue to the previous text mentioned by the authors in the manuscript.

"Therefore, it is possible to achieve better dyeing effects and improve the dyeing fastness properties by using a mixture of different types of dyes, which can help to overcome the limitations of natural yellow dyes."

Comment 3. This could be possible but, again, to an extent.

In my opinion, the author could describe more about the chemical behavior of yellow dyes, such as

(a) When mixed together, these dyes provide a synergistic effect that improves the overall dyeing of fabric or fiber. 

(b) chemically, most yellow dyes have phenolic groups, which help enhance the extent of chelation with metal ions (used as mordant or additive), and therefore, an improved dyeing can be observed.

In line 216, Result and Discussion, the Authors mentioned, "Plant dyes typically contain chromophores such as benzene rings or double bonds that can absorb UV or visible light and cause electron transitions"

Comment 4. The statement needs some attention and can be improved, especially the benzene ring and double bond.

For example, chromophores must contain a conjugate system that can absorb visible electromagnetic (EM) radiation in the UV-visible range, cause electron transition, or show absorption in the visible EM range.

These are suggestions, but authors are free to choose according to their interests.

Comment 5. In Figure 1 and Figure 6, the author showed structures of compounds that are dye molecules, however, their expression in the current extract was mentioned in the paper. 

There could be other compounds with yellow color profiles; how are these (shown in Figures 1 and 6) so specifically chosen for the current study? If authors haven't performed any experiments to evaluate the expression of these dye molecules in their extract, they could check previous publications where it has been done. Otherwise, the author needs to justify the expression of their presence or abundance in their extract.  

Author Response

Dear Mr. Zou, dear reviewers,

Thank you for your letter of April 28, regarding to your comments and questions to this manuscript. Firstly, we would like to thank you for the time and effort that you have put into the original version of the manuscript. Your suggestions have helped us greatly to improve our work. A revised version of the manuscript has been completed.

This letter contains our responses to the comments raised by the reviewers. These comments are in blue and our responses are given in black. We have tried our best to improve the manuscript. The changes were marked in red in the revised paper. However, the main content and framework of the paper haven’t been modified.

We sincerely thank you, editors and reviewers, for your enthusiastic work! We do hope that the revised manuscript will be approved and accepted in the Journal of Coatings.

Sincerely,

ZHAO Zhijun

  1. Authors mentioned, "Tannins have good dyeing fastness properties but lower dyeing color purity, while flavonoids (and alkaloids) have higher dyeing color purity but poor light-fastness properties."

Comment 1. What do authors mean by dyeing color purity? Are they indicating the labeling effect of dyeing OR the purity of dyes (in general)? Please state clearly.

Solution: dyeing color purity in this article refers to color purity of the dyed fabrics, which indicates the dyeing effect. The color purity of the dyed fabrics is represented by value C* (range from 0 to 100). The larger the value is, the purer the color is. We have made some modifications as followed.

Original text: Tannins have good dyeing fastness properties, but lower dyeing color purity. Flavonoids and alkaloids have higher dyeing color purity, but poor light-fastness properties.

Revised text: Tannin dyed fabrics have excellent fastness properties, but lower color purity. Flavonoid and alkaloid dyed fabrics have higher color purity, but poorer light fastness.

  1. "Dyes with similar physical and chemical properties show similar dyeing properties, color characteristics, and dyeing fastness"

Comment 2. This may be true but limited to only some cases, as substitution or modification of any particular position on chromophores or natural pigments can drastically change their color properties. Say example, naturally-derived anthraquinones are sensitive to any small modification on the ring, even so the position of substitution in the ring, and therefore any subtle change can alter the color and fastness properties.

Solution: thank you for raising this problem! We have made the following changes based on your suggestions.

Original text: Dyes with similar physical and chemical properties show similar dyeing properties, color characteristics, and dyeing fastness.

Revised text: Dyes with similar physical and chemical properties may show similar dyeing properties, color characteristics, and dyeing fastness.

  1. "Therefore, it is possible to achieve better dyeing effects and improve the dyeing fastness properties by using a mixture of different types of dyes, which can help to overcome the limitations of natural yellow dyes."

Comment 3. This could be possible but, again, to an extent.

In my opinion, the author could describe more about the chemical behavior of yellow dyes, such as

(a) When mixed together, these dyes provide a synergistic effect that improves the overall dyeing of fabric or fiber.

(b) chemically, most yellow dyes have phenolic groups, which help enhance the extent of chelation with metal ions (used as mordant or additive), and therefore, an improved dyeing can be observed.

Solution: thank you very much for your valuable comments! The original text is indeed lack of explanation. We have made some modifications based on your suggestions.

Original text: Therefore, it is possible to achieve better dyeing effects and improve the dyeing fastness properties by using a mixture of different types of dyes. which can help to overcome the limitations of natural yellow dyes.

Revised: Most of the yellow dyes contain phenolic groups, which can form coordination complexes with metal ions, such as aluminum and copper, and thus improve the color properties and the fastness of dyed fabrics.….. When applied together, some dyes may provide a synergistic effect, and improve the dyeing effect of the fabrics.

  1. In line 216, Result and Discussion, the Authors mentioned, "Plant dyes typically contain chromophores such as benzene rings or double bonds that can absorb UV or visible light and cause electron transitions"

Comment 4. The statement needs some attention and can be improved, especially the benzene ring and double bond.

For example, chromophores must contain a conjugate system that can absorb visible electromagnetic (EM) radiation in the UV-visible range, cause electron transition, or show absorption in the visible EM range.

These are suggestions, but authors are free to choose according to their interests.

Solution: Thank you very much for your valuable comments! Here we indeed need to improve our explanation. Based on your help we did following modifications.

Original text: Plant dyes typically contain chromophores such as benzene rings or double bonds that can absorb UV or visible light and cause electron transitions.

Revised: Plant dyes typically contain chromophores in their structure. Chromophores must contain a conjugate system that can absorb visible electromagnetic radiation in the UV-visible range, thus causing electron transition.

  1. These are suggestions, but authors are free to choose according to their interests.

Comment 5. In Figure 1 and Figure 6, the author showed structures of compounds that are dye molecules, however, their expression in the current extract was mentioned in the paper.

There could be other compounds with yellow color profiles; how are these (shown in Figures 1 and 6) so specifically chosen for the current study? If authors haven't performed any experiments to evaluate the expression of these dye molecules in their extract, they could check previous publications where it has been done. Otherwise, the author needs to justify the expression of their presence or abundance in their extract.

Solution: Thank you very much for your valuable comments! We haven’t done any quantitative analysis in this paper. The equipment needed in our school is under repair, and it is not available for a period of time. We would like to do an analysis of our extracts when the equipment is restored. Here we just did the literture review of the subject.

In the Research Progress on Chemical Constituents and Pharmacological Effects of Geranium Plants from Yang Yichen et al., it was shown that wild geranium contained these chemical components. And in the Simultaneous Determination of Seven Constituents in Three Kinds of Cranesbill by HPLC from Cao Bo et al., it was clearly pointed out that in the 3 varieties of wild geraniums, there were seven many components, which quantities were found in order of geraniin > Corilagin > Ellagic acid > Gallic acid > Protocatechuic acid > Hyperoside > Quercetin. In the Determination of Effective Components in Geranii herba from the Different Origins from Zhou Qingsong, it was pointed out that the content of gallic acid ranged from 0.047% to 0.294%, tannin from 1.08% to 2.02%, and total flavonoids from 0.87% to 1.83%. in the plants. These three references are listed below.

  • YANG Yi-chen ,CHANG Hui ,WANG Er-huan ,MA Cun-de ,LIU Feng ,WANG Ji-qiang ,JIN Peng-bo ,ZHAN Zhi-lai. Research Progress on Chemical Constituents and Pharmacological Effects of Geranium Plants [J]. Chinese Modern TCM, 2021,23(05):918-927.DOI:10.13313/j.issn.1673-4890.20200313003.
  • Cao Bo, Yin Haibo, Jia Xiaoqing, Shao Fei. Simultaneous determination of seven constituents in three kinds of cranesbill by HPLC.[J]. Chinese patent medicine, 2016,38 (06): 1338-1342.
  • ZHOU Qing song,SUN ï¼²uo fei,LI Xiao ran,YU Jie,DU Hua. Determination of effective components in Geranii herba from the different origins, 2015,30(04):465-468.DOI:10.13375/j.cnki.wcjps.2015.04.027.

Reviewer 2 Report

I examined this paper Ref. No. coatings-2350777 entitled “Study on Dyeing Properties and Color characteristics of Wool Fabrics dyed with Geranium caespitosum L. extract”.

This is a routine wool dyeing study with a new plant as a source of natural dye. However, in current format it is incomplete, yet there are some aspects and critical questions to be addressed to improve the quality of the paper. So, it is necessary authors to add some new tests and critically revise the manuscript. After that, revised manuscript may be re-examined for final decision. Comments are appended in the following  

Comments

Fig. 1 caption does not correlate with images

In current format, this study is incomplete. It is necessary to address following questions and issues well and revise the manuscript accordingly.

Using only one mordant (SnCl2) in dyeing with new natural dye is not sufficient as a new research paper. It is necessary to study different mordants (Al, Fe, Cu) and examine their performance on dyeing yield, coloring efficacy, mordanting method, color fastness, etc.   

Besides, ecological aspects and problems associated with use of metal mordants must be investigated. Tin salts pose serious ecological problems (the residual amount of Tin in the mordanting effluent, COD/BOD of both mordanting and dyeing effluent should be measured and reported).  

More in depth qualitative and quantitative analyses are needed to characterize and elucidate the phytochemicals and active coloring components of the plant like FTIR, Mass/Chromatography, etc.  

For dyed samples,

Please add K/S spectra of dyed samples and discuss the changes in shapes of spectra in accordance with mordant type, mordanting method, etc.

When using different metal mordants, please make use average color strength (K/S(avg)) instead of max. wavelength K/S value since it would give more accurate evaluation of dye up-take in all wavelengths.

SnCl2 is a reducing salt and damages wool structure and changes morphology. Please measure SEX-EDX of reference (raw fabric), mordanted and dyed mordanted samples.

It is stated that the plant possesses tannins. Tannins act as biomordants and improve the dyeing and fastness. What is the tannin content of this plant dye? Total tannins and total flavonoid content (quantitative method) of the dye should be measured and used in discussion of the results.

How is the reproducibility of the process investigated? How many tests for each sample have been done? Please add error bars for all figures and SD (standard deviation) values for all data.

The durability of color against successive laundering (0, 10, 20 wash cycles) and Xenon or UV-light aging is very important in real life and end-use application and assessment of the quality of the dyed product. Please measure and report the results and discuss the results accordingly.

Please add a cost analysis of the natural dyeing, comparing the eco-friendliness and sustainability with analogues chemical dyeing.

Some references used are unnecessary and should be omitted.

There are English language mistakes (word, grammar, adjective, and verbal) throughout the text.

Author Response

Dear Mr. Zou, dear reviewers,

Thank you for your letter of April 28, regarding to your comments and questions to this manuscript. Firstly, we would like to thank you for the time and effort that you have put into the original version of the manuscript. Your suggestions have helped us greatly to improve our work. A revised version of the manuscript has been completed.

This letter contains our responses to the comments raised by the reviewers. These comments are in blue and our responses are given in black. We have tried our best to improve the manuscript. The changes were marked in red in the revised paper. However, the main content and framework of the paper haven’t been modified.

We sincerely thank you, editors and reviewers, for your enthusiastic work! We do hope that the revised manuscript will be approved and accepted in the Journal of Coatings.

Sincerely,

ZHAO Zhijun

Reviewer 3 Report

The search for new functional solutions, including new dyes, has a justified practical purpose. In this regard, the work seems to be valuable. The introduction of the work is well developed, and the experimental part as well. The graphic part is aesthetically prepared, which makes the work a pleasure to read. The work seems complete and properly prepared for the publication process.

Certain aspects result from curiosity. Did the authors test the stained samples for leaching of the dye from the already colored product in terms of quantity? If so, it is worth mentioning them in the abstract.

line 126: g/m2

The English is fine.

Author Response

Dear Mr. Zou, dear reviewers,

Thank you for your letter of April 28, regarding to your comments and questions to this manuscript. Firstly, we would like to thank you for the time and effort that you have put into the original version of the manuscript. Your suggestions have helped us greatly to improve our work. A revised version of the manuscript has been completed.

This letter contains our responses to the comments raised by the reviewers. These comments are in blue and our responses are given in black. We have tried our best to improve the manuscript. The changes were marked in red in the revised paper. However, the main content and framework of the paper haven’t been modified.

We sincerely thank you, editors and reviewers, for your enthusiastic work! We do hope that the revised manuscript will be approved and accepted in the Journal of Coatings.

Sincerely,

ZHAO Zhijun

The search for new functional solutions, including new dyes, has a justified practical purpose. In this regard, the work seems to be valuable. The introduction of the work is well developed, and the experimental part as well. The graphic part is aesthetically prepared, which makes the work a pleasure to read. The work seems complete and properly prepared for the publication process.

  1. Certain aspects result from curiosity. Did the authors test the stained samples for leaching of the dye from the already colored product in terms of quantity? If so, it is worth mentioning them in the abstract.

Solution: Thank you very much for your suggestions.The washing fastness of the dyed fabrics has been measured and reached the standard.

Reviewer 4 Report

I quite like the idea of this paper, which is to investigate a new natural yellow dye and the application of that dye for coloring a variety of fabrics. Nonetheless, there are several minor issues (and a few more major ones) that combine to detract from my overall enthusiasm for this manuscript, and will need to be addressed before I can recommend publication. These issues include:

1.       This manuscript suffers from a non-trivial number of English language and syntax errors that make it difficult to focus exclusively on the scientific content of the manuscript. The authors should consider how to best address this issue.

2.       Much of the text of the introduction reads as hyperbolic and should be modified to more effectively convey scientific content. For example, the authors should consider modifying phrases such as “increasingly severe…” “exceed the capacity of natural balance,” and “most serious.”

3.       The authors list the potential sources of natural dyes, and include both ‘animals’ and ‘microorganisms.’ The distinction between these species should be made clear.

4.       In the introduction, the authors write that “natural dyes can provide more natural” colors. This is both redundant and hard to understand – what do the authors mean by ‘natural colors’?

5.       Similarly, the authors are advised to explicitly define what they mean by “natural” dyes.

6.       Also in the introduction, the authors seem to use the fact that traditional Chinese medicines have stable distributions as something that will benefit the natural dye industry. This connection is unclear and should be clarified by the authors.

7.       The authors seems to indicate that parts of Chinese medicines can be used as dyes….but this will almost certainly lead to significant deleterious effects in the ability of the Chinese medicinal industry to achieve therapeutic outcomes. If the goal is to divert materials from the Chinese medicine industry to that of natural dyes, then the costs to the medicinal industry need to be clearly elucidated.

8.       Also in the introduction, the authors state that “high-end clothing products” “must retain traditional ethnic colors.” This is hard to understand in the broad framework in which the authors are writing – i.e., given that there are many situations in which high end clothing products must have those ethnic colors, there are also at least an equal number of situations in which high end clothing is completely devoid from a particular ethnicity and need for ethnic colors. This phrasing and content should be modified accordingly.

9.       In the introduction, the authors refer to “metal salt mordants” such as “aluminum and copper.” This is imprecise. Aluminum and copper are elements, and metal salt mordants can contain salts of aluminum and copper, but the current phraseology is misleading and should be revised.

10.   The authors should define the terms of “dyeing fastness” and “dyeing color purity” the first time that they are used in the manuscript.

11.   The title of table 1 lists the dyes first, and then the chemical types, and does not list the plant sources at all. In contrast, the text of table 1 first lists the chemical type, then the examples of yellow dyes that fall into this category, and then the plant sources from which such dyes are derived. The authors are advised to modify either the title of Table 1 and/or the ordering of the columns in order to ensure consistency and clarity for the reader.

12.   The names of all plants (genus and species should be italicized in the text, not only in the table.

13.   The authors indicate that many ingredients from wild geranium contain “anti-ultraviolet” effects. This is nonsensical (i.e., what is an anti-ultraviolet effect?) and the authors should modify this phrase to more accurately convey their point.

14.   The caption for Figure 1 has nothing at all to do with the content of Figure 1

15.   Immediately after Figure 1, the authors write a paragraph about wool and keratin, which is not at all connected to any of the content that came before Figure 1. Assuming that the background information on wool is also relevant for this paper, the authors need to include a more direct link to the previous text of the introduction.

16.   In the paragraph on wool, the authors indicate that keratin molecules combine to form advanced structures, and list “disulfide bonds, ionic bonds, hydrogen bonds, and van der Waals forces” as types of advanced structures. None of the aforementioned items on the list constitute an “advanced structure.” The authors are encouraged to modify this sentence to more accurately reflect the scientific reality.

17.   In the text, the authors state that Figure 2 contains the UV visible spectroscopy results of a P. amurense dye solution. This is not at all related to what actually appears in Figure 2, nor is the P. amurense dye solution even a topic of this manuscript.  Corrections are needed.

18.   The authors do not define what the mordant process is, including what chemicals are involved, what temperatures, and what procedures. Without this information, it is impossible to understand the differences in the results that are obtained from fabrics that are subjected to the mordant process and those that are not.

19.   More significantly, the characterization of the extract from the G. caespitosum is severely underdeveloped. Merely reporting a UV spectrum of the extract and matching peaks to known standards is inconclusive at best. The authors need to conduct a much more comprehensive analysis of the extract in order to draw scientifically-based conclusions about the content of the extract. Only then can they comment to any significant extent on the relationship between the chemical structures of the compounds found in the extract and the observable properties of the dyes.

20.   Moreover, the introduction and much of the context of this paper focuses on a distinction between “natural” and “synthetic” dyes, and raises the value of the first class (i.e., the natural dyes) at the expense of the second class (i.e., the synthetic dyes). The authors do not consider the fact that extraction of natural food dyes can be both costly and environmentally damaging, they do not consider the fact that natural food dyes can have undesired impurities and/or toxicities that are easier to control in synthetic dyes, and they do not consider the long-term environmental cost of using such dyes. This lack of focus on the environmental costs of everything associated with natural food dyes makes it seem as if such dyes can solve all environmental challenges that currently exist! This is clearly untrue, and to the extent that the text of this manuscript leads a reader to that impression, the authors need to undertake a serious revision.

21.   Overall, this manuscript is far from ready for publication. The authors are advised to correct the careless errors (figure captions not matching figure content, incorrect names of dyes, etc.), conduct markedly more comprehensive characterization of every part of the chemical process, re-write the introduction to include a broader view of the field, clearly delineate the distinction between natural and synthetic dyes, and overall ensure that every part of the manuscript meets criteria of scientific accuracy and precision. Only then should they consider re-submission of the manuscript.

The authors are advised to consider how to address the nontrivial number of syntax-based and English language errors that detract from the ability to focus exclusively on the scientific content of the manuscript.

Author Response

(The authors gave the same response as above.)

Reviewer 5 Report

The authors presented “Study on Dyeing Properties and Color characteristics of Wool 2 Fabrics dyed with Geranium caespitosum L. extract” the work seems interesting; however the authors must address the following comments:

Please revise the title and make it specific

Please improve the abstract with respect to the objectives of the study

In description of Figure 1 it is described as molecular structure, however, in caption its written as UV spectrum! Please correct it!

Mostly figures are either wrongly captioned or un tagged. This must be addressed and aligned in the tet description  

The introduction section is very generalized. Please add the novelty of work and give comparison of natural dye with synthetic ones in accordance with recent literature

The presentation of the figures and their description is poor and not in sequence, so it must be addressed in organized manner.

The language of the article should be improved

Please double check the spelling and grammar issues

English should be improved

Author Response

Dear Mr. Zou, dear reviewers,

Thank you for your letter of May 6, regarding to your comments and questions to this manuscript. Firstly, we would like to thank you for the time and effort that you have put into the original version of the manuscript. Your suggestions have helped us greatly to improve our work. A revised version of the manuscript has been completed.

This letter contains our responses to the comments raised by the reviewers. These comments are in blue and our responses are given in black. We have tried our best to improve the manuscript. The changes were marked in red in the revised paper. However, the main content and framework of the paper haven’t been modified.

We sincerely thank you, editors and reviewers, for your enthusiastic work! We do hope that the revised manuscript will be approved and accepted in the Journal of Coatings.

Sincerely,

ZHAO Zhijun

Reviewer 6 Report

Comments for Editor and Authors

In the present the authors have extracted a new natural yellow dye from Geranium caespitosum L. Dyeing process of G. caespitosum dye on wool fabrics was optimized using single-factor experiments. Standard fastness tests were conducted to evaluate the sunlight, washing, and rubbing fastness of the dyed fabrics. The color characteristics and color gamut range of dyed fabrics were evaluated. The obtained colors were compared and assessed with Pantone Matching Systems and Chinese traditional colors. The experimental parameters are well planned and the results obtained were interpreted appropriately. In my opinion the manuscript may be accepted for publication in Coatings after minor revision. The specific comments are given below;

1.       In page 2, line 59, The reference 17 cannot be reached. Please check or provide a more appropriate reference.

2.       In page 3, the chemical formula of corilagin and gallic acid should be corrected as C27H24O18 and C7H6O5, respectively.

3.       In page 3, in line 89, the authors have given the molecular structures of ellagic acid, gallic acid, hyperoside, and quercetin in Figure 1. However the Fig 1 caption is “Figure 1. Absorption spectrum of P. amurense dye solution”. Please check and correct the figure caption.

4.       In page 3, in line 109, “Currently, there are many studies on the use of G. caespitosum extracts in the medical field.” Appropriate references are required for this statement.

5.       In page 3, line 113, K/S values should be written as “color strength (K/S) values”.

6.       In page 3, in line 117, “The UV–visible spectroscopy result of the P. amurense dye solution is shown below in Figure 2.”Please check the trueness of P. amurense dye. On the other hand the Fig 2 is related with Scheme of dyeing process, it not about The UV–visible spectroscopy result of dye. Please check and correct.

7.       In page 4, line 156, the authors used citric acid and sodium hydroxide as pH buffering agents. However they have carried out the experiments in the pH range of 4-11. What chemical reagents have the pH been adjusted in these ranges?

8.       The pH values should be given using two significant figures throughout the manuscript. For example pH 4 should be written as 4.0.

9.       “Sn ions” should be corrected as “Sn+2 ions” in whole manuscript.

 Comments for Editor and Authors

In the present the authors have extracted a new natural yellow dye from Geranium caespitosum L. Dyeing process of G. caespitosum dye on wool fabrics was optimized using single-factor experiments. Standard fastness tests were conducted to evaluate the sunlight, washing, and rubbing fastness of the dyed fabrics. The color characteristics and color gamut range of dyed fabrics were evaluated. The obtained colors were compared and assessed with Pantone Matching Systems and Chinese traditional colors. The experimental parameters are well planned and the results obtained were interpreted appropriately. In my opinion the manuscript may be accepted for publication in Coatings after minor revision. The specific comments are given below;

1.       In page 2, line 59, The reference 17 cannot be reached. Please check or provide a more appropriate reference.

2.       In page 3, the chemical formula of corilagin and gallic acid should be corrected as C27H24O18 and C7H6O5, respectively.

3.       In page 3, in line 89, the authors have given the molecular structures of ellagic acid, gallic acid, hyperoside, and quercetin in Figure 1. However the Fig 1 caption is “Figure 1. Absorption spectrum of P. amurense dye solution”. Please check and correct the figure caption.

4.       In page 3, in line 109, “Currently, there are many studies on the use of G. caespitosum extracts in the medical field.” Appropriate references are required for this statement.

5.       In page 3, line 113, K/S values should be written as “color strength (K/S) values”.

6.       In page 3, in line 117, “The UV–visible spectroscopy result of the P. amurense dye solution is shown below in Figure 2.”Please check the trueness of P. amurense dye. On the other hand the Fig 2 is related with Scheme of dyeing process, it not about The UV–visible spectroscopy result of dye. Please check and correct.

7.       In page 4, line 156, the authors used citric acid and sodium hydroxide as pH buffering agents. However they have carried out the experiments in the pH range of 4-11. What chemical reagents have the pH been adjusted in these ranges?

8.       The pH values should be given using two significant figures throughout the manuscript. For example pH 4 should be written as 4.0.

9.       “Sn ions” should be corrected as “Sn+2 ions” in whole manuscript.

Author Response

(The authors gave the same response as above.)

Round 2

Reviewer 4 Report

The authors have made some attempts to address many of the issues that I raised on the previous version of their manuscript. Nonetheless, there are still significant concerns that I have with the manuscript in its current form, and would recommend that the authors address these outstanding issues before I can recommend publication of the manuscript. The outstanding issues include:

1.       In the previous version of the manuscript, I stated that I thought the introduction had several statements that were hyperbolic and out of place in a scientific manuscript. Although the authors corrected the isolated examples of such language that I pointed out, the majority of the introduction still suffers from the same issue. I would suggest that the authors consider adding quantitative information to support their assertions, which is likely to render such assertions less hyperbolic. For example, instead of stating that “ecological and environmental problems” are “a great threat,” the authors should quantify what constitutes a “great threat.” Instead of stating that there is a “major global concern,” the authors should provide quantitative information to illustrate that the concern is in fact global, as they suggest. These are only two examples of statements that are written throughout the introduction, all of which should be clarified with quantitative metrics to support the assertions stated herein.

2.       On the previous version of the manuscript, I asked that the authors make the distinction between “animals” and “microorganisms” clear. The authors have responded to this request by adding parentheses after the word “animals,” to include a list of dyes that are derived from animals, and by adding parentheses after the word “microorganisms,” to include a list of species that are classified as “microorganisms.” This correction fails to address the issue. The authors should include both examples of dyes that are derived from each class of species (animals vs. microorganisms) as well as examples of living species that fall into each class (animals vs. microorganisms).

3.       On the previous version of the manuscript, the authors were asked to define what they mean when they say “natural” dyes. Instead of defining this adjective, they have instead provided two examples of such natural dyes – purple gum and carmine red. I will reiterate the request for a straightforward definition of “natural” dyes that is used by the authors throughout the manuscript and that will clearly distinguish these dyes from the “chemical synthetic dyes,” also referred to by the authors.

4.       Despite the fact that the authors claim that they have defined the terms “dyeing fastness” and “dyeing color purity,” no such definitions have been included in this revised manuscript.

5.       Despite the fact that the authors claim to have deleted the phrase “to form advanced structures” from their discussion about keratin in the introduction, and their stated claim that “disulfide bonds, ionic bonds, hydrogen bonds, and van der Waals forces” constitute such “advanced structures” – they have not in fact deleted this phrase. The sentence as written still states that the aforementioned list of intramolecular forces are examples of “advanced structures.” This is not correct and must be corrected.

6.       In the revised manuscript, the authors refer to “metal ions, such as aluminum and copper.” This is imprecise. As I already stated in my previous review, aluminum and copper are elements. They certainly can form metal cations, but the current phrasing of the text is inaccurate and misleading.

Overall, I would strongly recommend that the authors refrain from resubmitting a manuscript until and unless they have addressed all points raised by the reviewers. While they are under no obligation to accept all of the reviewer’s comments/ suggestions, they are advised to strongly consider each point, and respond in a thoughtful and scientifically accurate way. The current revision of the manuscript, submitted by the authors in response to my previous comments on their manuscript, falls far short of the objective of responding to reviewer comments in a “thoughtful and scientifically accurate way.”

The English language is certainly improved from the previous version of the manuscript, although additional language edits/ proofreading are still recommended.

Author Response

Dear Mr. Zou, dear reviewers,

Thank you again for your detailed and constructive suggestions and comments on our manuscript! We have further worked on these points and made some modifications. We have tried our best to improve the quality of our manuscript and hope this revised version can reach the publican standard. Please kindly find our solutions of your comments marked in blue.

We would like to thank you again, dear reviewer, for your great help! We do hope that this revised manuscript would be approved and accepted in the Journal of Coatings.

Sincerely,

ZHAO Zhijun

Round 3

Reviewer 4 Report

While the authors have done a reasonable job in responding to my comments and suggestions on the second version of their manuscript, there remain a few unresolved issues that should be addressed before I can recommend publication of the manuscript. These include:

1.       In the introduction, I have suggested that the authors provide quantitative data to support what would otherwise sound like hyperbolic statements. I provided two examples of situations in which quantitative data would be helpful, and stated clearly that these are only two of several situations in which quantitative data would improve the scientific content of the introduction. Instead of addressing this issue globally, the authors have decided to simply address the two examples that I raised. This response raises additional questions as to the seriousness of the authors in addressing reviewer comments, particularly through multiple revision cycles.

2.       I have asked more than once for the terms ‘dyeing fastness’ and ‘dyeing color purity’ to be defined. The authors have elected to define ‘dyeing fastness,’ but have not defined ‘dyeing color purity,’ despite my repeated requests.

Once these outstanding issues have been addressed, I would be pleased to recommend that the manuscript be accepted for publication.

Author Response

Dear reviewer,

Thank you again for your detailed and constructive suggestions and comments on our manuscript! We have tried our best to improve. Please kindly find our solutions of your comments marked in blue.

We would like to thank you again, dear reviewer, for your great help! We do hope that this revised manuscript would be approved and accepted in the Journal of Coatings.

Sincerely,

ZHAO Zhijun

While the authors have done a reasonable job in responding to my comments and suggestions on the second version of their manuscript, there remain a few unresolved issues that should be addressed before I can recommend publication of the manuscript. These include:

  1. In the introduction, I have suggested that the authors provide quantitative data to support what would otherwise sound like hyperbolic statements. I provided two examples of situations in which quantitative data would be helpful, and stated clearly that these are only two of several situations in which quantitative data would improve the scientific content of the introduction. Instead of addressing this issue globally, the authors have decided to simply address the two examples that I raised. This response raises additional questions as to the seriousness of the authors in addressing reviewer comments, particularly through multiple revision cycles.

Solution: We have modified the first two paragraphs of the article as below.

The first paragraph of the previous manuscript:

The ecological and environmental issues caused by industrial development have always been present. Particularly, chemical synthetic dyes are widely used in many industries such as textiles, food, cosmetics, and pharmaceuticals [1, 2]. In the textile industry, 1.3 million tons of synthetic dyes are used for textile dyeing and printing worldwide each year, and the discharge of dye wastewater into freshwater ecosystems affects the quality parameters of freshwater, such as pH, turbidity, color, chemical oxygen demand, heavy metal content, organic matter content, etc., posing a certain threat to humans and the environment [3-5]. As the restrictions on the use of synthetic dyes gradually increase [6], and with the widespread adoption of the green and sustainable development concept, more researchers are focusing on the study of natural dyes.

Revised version:

The ecological and environmental issues caused by industrial development have always been a concern. It has been estimated that over 700 newly identified pollutants, such as waste from petrochemicals, personal care products, textiles, and pesticides, have been confirmed in aquatic ecosystems in the European region. Among them, the textile and dyeing industry is considered a major source of water pollution [1]. Synthetic dyes, in particular, are widely used in various sectors such as textiles, food, cosmetics, and pharmaceuticals [2-3]. In the textile industry alone, about 1.3 million tons of synthetic dyes are used for textile printing and dyeing annually worldwide, with approximately 8-20% of unused dyes and auxiliary chemicals discharged into wastewater [4-5]. Dye wastewater contains high levels of biochemical oxygen demand (BOD) and chemical oxygen demand (COD) [6-7]. When dye wastewater is discharged into freshwater ecosystems, it affects the quality parameters of freshwater, such as pH, turbidity, color, and chemical oxygen demand etc. Discharging contaminated wastewater into the environment without any treatment poses various environmental threats, including inhibition of photosynthesis and death of aquatic plants [8]. As the restrictions on the use of synthetic dyes gradually increase [9], and with the widespread adoption of the green and sustainable development concept, more researchers are focusing on the study of natural dyes.

  • Li, Y.; Ren, J.; Chen, S.; Fan, F.; Shen, Q.; Wang, C., Cationic superfine pigment dyeing for wool using exhaust process by pH adjustment. Fibers Polymers 2015, 16, (1), 67.^https://doi.org/10.1007/s12221-015-0067-0.
  • Li, N.; Wang, Q.; Zhou, J.; Li, S.; Liu, J.; Chen, H., Insight into the Progress on Natural Dyes: Sources, Structural Features, Health Effects, Challenges, and Potential. Molecules 2022, 27, (10), 3291.^ https://doi.org/10.3390/molecules27103291.
  • Adeel, S.; Zia, K. M.; Abdullah, M.; Rehman, F.-u.-.; Salman, M.; Zuber, M., Ultrasonic assisted improved extraction and dyeing of mordanted silk fabric using neem bark as source of natural colourant. Natural product research 2019, 33, (14), 2060-2072.^https://doi.org/10.1080/14786419.2018.1484466.
  • Adeel, S.; Salman, M.; Usama, M.; Rehman, F.-u.-.; Ahmad, T.; Amin, N., Sustainable Isolation and Application of Rose Petals Based Anthocyanin Natural Dye for Coloration of Bio-Mordanted Wool Fabric: Short title: Dyeing of Bio Mordanted Wool With Rose Petal Extract. Journal of Natural Fibers 2022, 19, (13), 6089-6103.^https://doi.org/10.1080/15440478.2021.1904480.
  • Uday, U. S. P.; Bandyopadhyay, T. K.; Bhunia, B., Bioremediation and detoxification technology for treatment of dye (s) from textile effluent. InTech London: 2016; Vol. 4.
  • Bansal, P.; Sud, D., Photodegradation of commercial dye, CI Reactive Blue 160 using ZnO nanopowder: Degradation pathway and identification of intermediates by GC/MS. Separation Purification Technology 2012, 85, 112-119.^https://doi.org/10.1016/j.seppur.2011.09.055.
  • Champagne, P.-P.; Nesheim, M.; Ramsay, J., Effect of a non-ionic surfactant, Merpol, on dye decolorization of Reactive blue 19 by laccase. Enzyme microbial technology 2010, 46, (2), 147-152.^https://doi.org/10.1016/j.enzmictec.2009.10.006.
  • Mansoorian, H. J.; Bazrafshan, E.; Yari, A.; Alizadeh, M., Removal of azo dyes from aqueous solution using Fenton and modified Fenton processes. Health Scope 2014, 3, (2).^https://doi.org/10.17795/jhealthscope-15507.
  • Wani, S. A.; Mohammad, F., Imparting functionality viz color, antioxidant and antibacterial properties to develop multifunctional wool with Tectona grandis leaves extract using reflectance spectroscopy. International journal of biological macromolecules 2018, 109, 907-913.^https://doi.org/10.1016/j.ijbiomac.2017.11.068.

In the second paragraph, we modified the first several sentences:

The second paragraph of the previous manuscript:

Natural dyes are dyes extracted from certain parts of plants, animals, minerals, and microorganisms. They are commonly found in plant flowers, fruits, leaves, roots, and stems, as well as in animal secretions and bacterial strains [7]. Carminic acid, a natural animal red dye, is a quinone pigment obtained from the cochineal insect, which can dye wool fabric a purplish-red color [8]. The lac resin, secreted by the lac insect, is also an animal dye that can dye fabrics purple or red [9]. Prodigiosin, extracted from the bacterium Serratia marcescens, is a microbial dye that can be used for textile dyeing [10]. Fusarium oxysporum, a fungus, can produce pink-purple anthraquinone pigments, which are also microbial dyes used for dyeing wool fabrics, giving them vivid colors [11]. Among them, plant dyes have received significant attention due to their advantages such as being healthy and safe, biodegradable, environmentally compatible, and their therapeutic functionalities, which meet the requirements of developing eco-friendly textiles and sustainable concepts [5, 12].

Revised version:

Natural dyes are dyes extracted from certain parts of animals, plants, minerals, and microorganisms. They are commonly found in the juices of plant flowers, fruits, leaves, roots, and stems, as well as in animal secretions and bacterial strains [10]. Natural dyes extracted from these sources have several advantages: they are biodegradable [11], renewable [12], and environmentally compatible, meaning they do not disrupt ecosystems. Carminic acid, a natural animal red dye, is a quinone pigment obtained from the cochineal insect, which can dye wool fabric a purplish-red color [13]. The lac resin, secreted by the lac insect, is also an animal dye that can dye fabrics purple or red [14]. Prodigiosin, extracted from the bacterium Serratia marcescens, is a microbial dye that can be used for textile dyeing [15]. Fusarium oxysporum, a fungus, can produce pink-purple anthraquinone pigments, which are also microbial dyes used for dyeing wool fabrics, giving them vivid colors [16]. Among them, plant dyes have received significant attention due to their advantages such as being healthy and safe, biodegradable, environmentally compatible, and their therapeutic functionalities, which meet the requirements of developing eco-friendly textiles and sustainable concepts [4, 17].

  • Adeel, S.; Rehman, F.-U.-.; Zia, K. M.; Azeem, M.; Qayyum, M. A., Microwave-Supported Green Dyeing of Mordanted Wool Fabric with Arjun Bark Extracts. Journal of Natural Fibers 2019, (1), 1-15.^https://doi.org/10.1080/15440478.2019.1612810.
  • Kalsy, M.; Srivastava, S. In Dyeing of silk with RosaCentifolia: An Eco-Friendly approach, International conference on interdisciplinary. Research in Engineering andTechnology, 2016; p 46.
  • Karaboyaci, M., Recycling of rose wastes for use in natural plant dye and industrial applications. The journal of the textile institute 2014, 105, (11), 1160-1166.^https://doi.org/10.1080/00405000.2013.876153.
  • Adeel, S.; Hussaan, M.; Rehman, F. U.; Habib, N.; Salman, M.; Naz, S.; Amin, N.; Akhtar, N., Microwave-assisted sustainable dyeing of wool fabric using cochineal-based carminic acid as natural colorant. Journal of Natural Fibers 2018, 1-9.^https://doi.org/10.1080/15440478.2018.1448317.
  • SONG Huijun, D. L., CAO Keke, Research of lac pigment dyeing silk fabric Textile Dyeing and Finishing Journal 2017, 39, 18-22.^https://doi.org/1005-9350(2017)12-0018-05.
  • , M. R.; Amany, E. S.; A., E. S. N.; A., G. H.; A., S. S., Antimicrobial activity of textile fabrics dyed with prodigiosin pigment extracted from marine Serratia rubidaea RAM_Alex bacteria The Egyptian Journal of Aquatic Research 2021, 47, (3).^https://doi.org/10.1016/j.ejar.2021.05.004.
  • Nagia, F. A.; El-Mohamedy, R., Dyeing of wool with natural anthraquinone dyes from Fusarium oxysporum. Dyes Pigments 2007, 75, (3), 550-555.^https://doi.org/10.1016/j.dyepig.2006.07.002.
  • Zhang, W.; Wang, X.; Weng, J.; Liu, X.; Qin, S.; Li, X.; Gong, J., Eco-dyeing and functional finishing of wool fabric based on Portulaca oleracea L. as colorant and Musa basjoo as natural mordant. Arabian Journal of Chemistry 2022, 15, (2), 103624.^https://doi.org/10.1016/j.arabjc.2021.103624.

  1. I have asked more than once for the terms ‘dyeing fastness’ and ‘dyeing color purity’ to be defined. The authors have elected to define ‘dyeing fastness,’ but have not defined ‘dyeing color purity,’ despite my repeated requests.

Solution: thank you for pointing out again this problem! However, we have added the definition of dyeing color purity in the previous manuscript. Here we again did some modifications and made it clearer. Please kindly find out the following changes.

Original text:

However, each type of yellow dye has its defects. For example, tannin-based yellow dyes have excellent color fastness properties but lower color purity. Flavonoid and alkaloid dyed fabrics have higher color purity, but poorer light fastness [27, 28]. The purity or intensity of a color refers to the chroma or saturation of a color. A high chroma or saturation indicates a vivid, pure, and vibrant color, while a low chroma or saturation results in a more subdued or desaturated color appearance [29].

Revised version:

However, each type of yellow dye has its defects. For example, tannin-based yellow dyes have excellent color fastness properties but lower color purity. Flavonoid and alkaloid dyed fabrics have higher color purity, but poorer light fastness [32, 33]. The dyeing purity or intensity of a color refers to the chroma or saturation of a color, which is an important indicator for evaluating color. It can be measured by a colorimeter, which is represented by the parameter C*, with a range from 0 to 100. A high C* value indicates a vivid, pure, and vibrant color, while a low C* value results in a more subdued or desaturated color appearance [34].

[32]. Vankar, P. S.; Shukla, D., Spectrum of colors from reseda luteola and other natural yellow dyes. Textile Eng. Fashion Technol 2018, 4, (2), 107-120.^https://doi.org/10.15406/jteft.2018.04.00127.

[33].Lee, Y. H.; Hwang, E. K.; Jung, Y. J.; Do, S. K.; Kim, H. D., Dyeing and deodorizing properties of cotton, silk, wool fabrics dyed with Amur Corktree, Dryopteris crassirhizoma, Chrysanthemum boreale, Artemisia extracts. Journal of Applied Polymer Science 2010, 115, (4), 2246-2253.^ https://doi.org/10.1002/app.31357.

[34].Wang Chuandong , W. B., Photography and Camera Shooting. 2014.